# FPGA-Based Hardware Implementation of a Stable Inverse Source Problem Algorithm in a Non-Homogeneous Circular Region

José Jacobo Oliveros-Oliveros [1,*,†], José Rubén Conde-Sánchez [1,†], Carlos Arturo Hernández-Gracidas [2,†], María Monserrat Morín-Castillo [3,†] and José Julio Conde-Mones [1,†]

1    Facultad de Ciencias Físico Matemáticas, Benemérita Universidad Autónoma de Puebla, Avenida San Claudio y 18 sur, Colonia San Manuel, Edificio FM1-101B, Ciudad Universitaria, Puebla C.P. 72570, Mexico; rconde@fcfm.buap.mx (J.R.C.-S.); jose.conde@correo.buap.mx (J.J.C.-M.)
2    CONAHCYT-BUAP, Facultad de Ciencias Físico Matemáticas, Benemérita Universidad Autónoma de Puebla, Avenida San Claudio y 18 Sur, Colonia San Manuel, Edificio FM1-101B, Ciudad Universitaria, Puebla C.P. 72570, Mexico; cahernandezgr@conahcyt.mx
3    Facultad de Ciencias de la Electrónica, Benemérita Universidad Autónoma de Puebla, Avenida San Claudio y 18 Sur, Colonia San Manuel, Edificio FCE1, Ciudad Universitaria, Puebla C.P. 72570, Mexico; maria.morin@correo.buap.mx
*    Correspondence: jose.oliveros@correo.buap.mx; Tel.: +52-(222)-2295500 (ext. 2178)
†    These authors contributed equally to this work.

**Abstract:** Objective: This work presents an implementation of a stable algorithm that recovers sources located at the boundary separating two homogeneous media in field-programmable gate arrays. Two loop unrolling architectures were developed and analyzed for this purpose. This inverse source problem is ill-posed due to numerical instability, i.e., small errors in the measurement can produce significant changes in the source location. Methodology: To handle the numerical instability when recovering these sources, the Tikhonov regularization method in combination with the Fourier series truncation method are applied in the stable algorithm. This stable algorithm is implemented in two different architectures developed in this work: The first architecture (Mode 1) allows for different operating speeds, which is an advantage depending on whether we work with fast or slow signals. The second one (Mode 2) reduces resource consumption by exploiting the characteristics of the source identification algorithm, which is an advantage for multichannel problems such as inverse electrocardiography or electroencephalography. Results: The architectures were tested on four devices of the 7 Series of Xilinx: Spartan-7 xc7s100fgga484, Virtex-7 xc7v585tffg1157, Kintex-7 xc7k70tfbg484, and Artix-7 xc7a35tcpg236. The two hardware implementations of the stable algorithm were validated using synthetic examples implemented in MATLAB, which shows the advantages of each architecture. Contributions: We developed two efficient architectures based on a loop unrolling design for source identification problems. These are effective strategies to divide and assign tasks to the configurable hardware, and they appear as an appropriate technique for implementing the algorithm. The first one is simple and allows for different operating speeds. The second one uses a control system based on multiplexors that reduce resource consumption and complexity of the design and can be used for multichannel problems. From the numerical test, we found the regularization parameters. The synthetic examples developed here can be considered for similar problems and can be extended to concentric spheres.

**Keywords:** field-programmable gate arrays (FPGAs); loop unrolling architectures; inverse source problem; ill-posed problems; Tikhonov regularization

## 1. Introduction

In this paper, we propose two architectures based on field-programmable gate arrays (FPGAs) [1] for implementing an inverse source identification algorithm. The inverse

source identification problem that we address involves determining the source from measurements produced by it on the exterior boundary. The inverse source problem has many applications in practical problems such as inverse electroencephalography and inverse electrocardiography, for which study has been conducted using operational equations defined in Hilbert spaces [2,3]. These operational equations are ill-posed in the sense of Hadamard [4]. Here, we solved the operational equation, from which, the algorithm was obtained using Fourier series. More precisely, we solved the normal equations obtained using the Tikhonov regularization method. Some inverse source problems lead us to algebraic systems of linear equations, which are obtained from the operational equations by discretization [5]. The matrices in these systems of equations are ill-conditioned, which must be considered when we solve the system with error on the right side because the systems are unstable, which is a consequence of the ill-posedness of the operational equations. At this point, it is important to consider that handling precision inadequately might worsen the results, as the solution found for data with errors might be too far from the one for data without errors in addition to increasing other costs via hardware resources and critical paths. In [6], the authors pointed out that using $2 \times 2$ matrices to show how ill-conditioning and precision can affect system design (resources, cost, etc.), and they illustrated the effect generated in the calculation of the inverse of an ill-conditioned matrix when its elements were approximated by truncation.

We consider a circular, non-homogeneous medium and that the sources are located on the separation interface of two homogeneous media, which make up the non-homogeneous media. Sources and measurements are correlated by a boundary value problem, which allows us to make an operational statement from where the ill-posedness (in the sense of Hadamard) is analyzed, and the source identification algorithm is obtained. We consider the algorithm given in [7], in which the inverse source problem was developed for sources located on the interface of two homogeneous media. The authors developed the algorithm using the technique of Fourier series for circular geometry and the finite element method for a complex geometry. Since we are considering a circular geometry, we implemented the algorithm in an FPGA using the Fourier series technique, i.e., we used the trigonometric base to express the solution to the inverse problem. Ill-posedness is related to numerical instability, which can produce significant changes in the solution due to small measurement changes. The Tikhonov regularization method is employed to handle this numerical instability; it depends on a parameter called the Tikhonov regularization parameter, which must be properly chosen in terms of the measurement error [5]. We chose the Tikhonov regularization parameter numerically.

Associated with the trigonometric base is an arithmetic kernel, which allows hardware resources to be reused in a repetitive type system to reduce the number of operations [8]. The proposed architectures, labeled Mode 1 and Mode 2, were tested on four FPGAs of the 7 Series of Xilinx [1]: Spartan-7 xc7s100fgga484, Virtex-7 xc7v585tffg1157, Kintex-7 xc7k70tfbg484, and Artix-7 xc7a35tcpg236. We report on the performance of architectures Mode 1 and Mode 2 in terms of power and resource consumption.

The two architectures can also be replicated, leading to a multichannel structure. This offers the possibility of applying these architectures to problems for which their nature is multichannel, such as identifying bioelectrical sources from electroencephalographic or electrocardiographic signals. In the first case, voltage measurements are taken on the scalp through electrodes following different arrangements (10–20 being the most-used). Up to a thousand measurements per second are recorded on each electrode, which implies applying the source identification algorithm the same number of times. In the case of electroencephalography, for which the maximum frequency is 120 Hz, it is feasible to use both architectures. However, the Mode 2 architecture consumes fewer hardware resources, which can contribute to the development of portable electroencephalographs.

The problem that we are addressing involves developing and implementing two efficient loop unrolling architectures for a stable source identification algorithm in FPGAs.

To achieve this, we set the following objectives:

1.  Develop loop unrolling architectures for implementing a stable source identification algorithm.
2.  Analyze the stable algorithm to determine the regularization parameters numerically.
3.  Validate the FPGA implementations.

We make the following contributions:

1.  Development of two efficient architectures based on a loop unrolling design for source identification problems. Architecture Mode 1 is simple and allows for different operating speeds.
2.  Architecture Mode 2 uses a control system based on multiplexors, reducing resource consumption and complexity and allowing multichannel problem usage.
3.  Implementation of the source identification algorithm in four FPGAs.
4.  Design of a control system based on multiplexors for both architectures.
5.  Analysis and comparison of resource consumption.
6.  Creation of ad hoc synthetic examples: particularly, the jump function, which is commonly used in electronic applications.
7.  Implementation of synthetic examples in MATLAB 2013a.
8.  Numerical analysis to choose the regularization parameters to handle the numerical instability of the algorithm.
9.  Validation of the performance of the hardware architectures compared with software results in terms of error.

This work is organized as follows. In Section 2, we present the foundations for this work. Section 3 presents the mathematical model that relates the sources with measurements. Furthermore, we solve the forward and inverse problems. The solution to the inverse problem provides the stable source identification algorithm; Section 4 presents numerical examples associated with forward and inverse problems. These examples are developed using MATLAB software to validate the source identification algorithm given in Section 3. In Section 5, we implement the source identification algorithm in FPGAs using the two proposed architectures. In Section 6, we validate the hardware implementations using the same examples developed in Section 4. In Section 7, we discuss the obtained results. Finally, we present our conclusions in Section 8.

## 2. Basic Elements

This section presents the foundational components of the research. It serves as the cornerstone for the rest of the paper and provides the reader with the necessary background information to understand the main results and conclusions.

### 2.1. FPGA

An FPGA is an integrated circuit designed to be configured after the manufacture, assembly, and even deployment of the product it is part of. FPGAs contain programmable logic blocks (capable of performing a range of operations from simple logic gates to complex functions) and memory blocks. These blocks can then be connected using a hierarchy of reconfigurable interconnects. The combination of these elements improves the performance of the FPGA in many varied applications.

Some advantages of FPGAs are:

1.  FPGAs are more flexible than complex programmable logic devices as they generally have a greater number of both logic blocks and programmable interconnects.
2.  FPGAs have a lower development cost than application-specific integrated circuits (ASICs). Although an ASIC can perform the same operations as an FPGA and is specific to the application, it cannot be reprogrammed.
3.  FPGAs have a faster time-to-market and lower non-recurring engineering cost than ASICs.

Three of the main characteristics of FPGAs, which are related to the speed of the process, are:

1.  Throughput: the amount of data that are processed per clock cycle (bits/second).
2.  Latency: the time between data input and processed data output (clock cycles).
3.  Timing: the logic delays between sequential elements (frequency).

### 2.2. Inverse and Ill-Posed Problems

On the one hand, inverse problems entail finding an unknown property of an object or medium from observations of its responses to test signals. Forward problems, on the other hand, present information about the causes that describe a process in a medium, and the solution to the problem leads to the discovery of the effect produced by such causes. Therefore, in contrast to forward problems, inverse problems provide partial information about the results or effects produced in the medium by some unknown causes that must be found by analyzing these results. Thus, forward problems are cause–effect problems, while inverse problems are effect–cause problems [4].

Source identification problems are widely investigated in many research fields, and they are modeled as boundary value problems for which both the associated forward problem and its corresponding inverse problem must be considered. The inverse problem involves determining the source that produced measurements on the boundary of a region. It appears in applications such as inverse electroencephalography, inverse electrocardiography, and inverse geophysics, for which problems are modeled using partial differential equations.

An operational equation of the form $Ax = y$, where $A : X \longrightarrow Y$ and $X$ and $Y$ are Banach or Hilbert spaces, is well-posed if it satisfies the following conditions [4]:

1.  For each $y \in Y$, a solution to the problem exists.
2.  For each $y \in Y$, the solution to the problem is unique.
3.  The solution $x$ to the problem continuously depends on initial data $y$.

Problems that violate one or more of these conditions are referred to as ill-posed problems. Condition 3 is associated with numerical instability, i.e., small errors on the right side of the operational equation can result in significant changes to the location of the solution $x$. To address this numerical instability, regularization methods must be applied. In this work, we applied the Tikhonov regularization method, which involves choosing a regularization parameter in terms of the error [4,5].

### 2.3. FPGAs and Inverse Source Problems

Inverse source problems appear in many applications. One such application is the inverse electroencephalographic problem, which involves determining bioelectrical sources of the brain from scalp measurements of the electrical potential produced by these sources. Identifying dipolar sources associated with epileptic foci is particularly important given that epilepsy affects around 50 million people worldwide according to the World Health Organization. Another application is the inverse electrocardiography problem, which involves determining the epicardial potential from measurements of the potential on the chest. These examples are multichannel problems for which the architectures developed in this work can be applied. These are generally ill-posed problems in the sense of Hadamard due to the non-uniqueness of the solution and the numerical instability, which can produce large changes to the location of the source when errors are presented. Regularization methods are employed to handle this instability. The Tikhonov regularization method is the best known among them; it depends on a parameter that must be chosen in terms of the measurement error. The identification problem studied in this work only presents numerical instability since the problem has a unique solution [7,9]. The regularization in this work considers two parameters. The former is the Tikhonov regularization parameter. The latter is the term in which the series is truncated. From numerical tests, we found these parameters. This is a contribution of this work.

FPGA devices offer several advantages for the implementation of algorithms: the primary one being their capability to accelerate algorithms. Inverse source identification problems, which appear in many applications, are one such area where these advantages

can be leveraged. In this work, we have developed two architectures for an algorithm designed to solve inverse source identification problems.

A natural question arises: How are these problems and the architectures related? The answer lies in the stable source identification algorithms (one for each of them). This work presents two roll-up architectures in which a stable algorithm was implemented to recover sources located on the interface that separates two homogeneous media that make up a non-homogeneous medium. The algorithm is developed in the bidimensional case, but the tridimensional case corresponds to the identification of sources located in the cerebral cortex. In the tridimensional case, we must use other elements for the base (spherical harmonics instead of circular harmonics), but the methodology presented here can be extended without problem.

One of the key advantages of the architectures developed in this work is our ability to determine the number of coefficients required, as this number significantly influences the numerical stability of the algorithm. Additionally, we can select the number of bits to achieve accurate approximations while minimizing hardware consumption. For instance, the tridimensional version of the algorithm is presented here. These developments can assist us with constructing portable devices for source and potential identification. Furthermore, they can facilitate the design and implementation of reconfigurable devices for real-time imaging of sources.

It is crucial to highlight that these architectures are designed to serve as a foundation for other algorithms. We can deduce that the architectures developed in this work can be repurposed for other similar algorithms by leveraging their reconfigurable features. In the following, we provide some insights into FPGAs and the loop unrolling architectures.

Broadly speaking, the design of systems based on FPGAs is geared towards enhancing performance and optimizing loops. The loop unrolling technique, which expands loops into an iterative version, has shown promising results in this area [10]. Loop unrolling architectures are effective strategies for distributing tasks to configurable hardware, reducing overhead, and enabling parallel processing [11,12]. Moreover, loop unrolling is advantageous as it minimizes latency and optimizes the use of hardware resources.

In Section 2.4, we detail several algorithms implemented on FPGAs. These algorithms are related to the Fourier transform, the resolution of systems of algebraic equations, finite impulse response filters, and the classification of generalized and focal epileptic seizure types. Despite the advantages of loop unrolling architectures, our comprehensive literature review did not uncover any instances for which loop unrolling was employed to implement source identification algorithms.

In light of this, we have implemented a stable algorithm in two loop unrolling architectures that were developed as part of this work.

### 2.4. Implementation of Algorithms in FPGAs

A variety of algorithms have been implemented in FPGAs [13–18]. The study [16] addresses the challenge of implementing histogram projection using FPGAs. Histogram projection is an effective enhancement technique for images captured by uncooled infrared imagers. Thermal imagers that utilize uncooled focal plane array detectors offer several significant advantages over their cooled counterparts. These advantages include lower cost, silent operation, absence of delicate mechanical parts for cooling, reduced lifecycle cost, and decreased weight and power consumption. Despite suffering from limitations in thermal and geometrical resolutions, these imagers are highly performant and are widely accepted for commercial applications. This acceptance is largely due to their reasonable price point, especially when compared to cooled imagers, which, while offering superior image quality, are prohibitively expensive for commercial use.

In the study [17], digital watermarking is examined. This technique, which embeds watermark information into a digital signal in a way that makes it difficult to remove, is of significant importance. The authors developed an adaptive digital watermarking algorithm to improve performance in a multi-parametric solution space for hiding copyright

information. This was achieved through the use of phase congruence and singular-value decomposition, which was supported by an information-hiding technique. The Tikhonov regularization method is analyzed as a filter that reduces the effect of the small-singular-value decomposition using the regularization parameter. The truncation of the series is considered as another regularization parameter: thereby eliminating the effects of the small-singular-value decomposition. The performance of the algorithm is evaluated through a simulation in MATLAB using metrics such as the peak signal–noise ratio, structural similarity index metrics, and the normalized cross-correlation index.

In [13], the authors present an efficient FPGA implementation of a reconfigurable finite impulse response (RFIR) filter. This implementation utilizes anti-symmetric product coding and odd multiple storage modules. The finite impulse response filter is widely used in various digital signal processing applications, such as echo removal, speech signal processing, speaker standardization, versatile noise removal, and communication. The RFIR filter has the advantage of allowing real-time changes to the coefficient while performing operations. In this particular case, filters were not used to clean the signal. Instead, a random error was emulated by adding it using the *rand* function of MATLAB.

In the study [18], the authors developed an FPGA-based solution for classifying generalized and focal epileptic seizure types using a feed-forward multi-layer neural network architecture. The FPGA implementation in the last two works was validated by comparing its results with those of the MATLAB implementation: a method also employed in our study to validate our FPGA implementation.

In [14], the authors implemented the minimum residual algorithm: an effective method for solving problems provided the matrix exhibits certain characteristics. The paper examines an IEEE 754 single-precision floating-point implementation of the algorithm on an FPGA. It demonstrates that through parallelization and heavy pipelining of all floating-point components, it is possible to achieve sustained performance of up to 53 GFLOPS on the Virtex5-330T. This performance compares favorably to other hardware implementations of floating-point matrix inversion algorithms and represents an improvement of nearly an order of magnitude compared to a software implementation.

In [15], the authors present an efficient implementation of a multiplier of irreducible polynomials modulo for cryptographic encryption and decryption using FPGAs. For this purpose, the Nexys 4 board based on the Artix-7 FPGA from Xilinx was chosen. Verilog HDL was used to describe the circuit for reducing a number modulo. The results of a timing simulation of the device are presented in the form of time diagrams for a given 8-bit number and confirm the correct operation of the device.

In [19], the authors presented the first implementation of a 1 million point fast Fourier transform completely integrated on a single FPGA without the use of external memory or multiple interconnected FPGAs. The proposed architecture is a pipelined single-delay feedback fast Fourier transform, which allows for processing one sample per clock cycle.

Loop unrolling architectures have been utilized to implement the algorithms described in this section. Some of these algorithms correspond to Fourier transforms, the solution to systems of algebraic equations, RFIR filters, the classification of generalized and focal epileptic seizure types, and adaptive digital watermarking. According to our search, we did not find any implementations related to the inverse source algorithm. As previously mentioned, the inverse source problem that we study in this paper is ill-posed because it presents numerical instability, making it a challenging task to implement on hardware. To address this issue, we utilized the Tikhonov regularization method and the truncation method. To the best of our knowledge, this work is the first to present the hardware implementation of two architectures for a stable source identification algorithm on FPGAs. Our approach demonstrates that the algorithm can be parallelized to take advantage of the unique capabilities of FPGAs. This implementation demonstrates the feasibility and efficiency of utilizing FPGAs for inverse source problems, which is an important and growing field [2,3,9,20,21]. Hence, the implementation developed in this work provides a new and promising solution for future research in this field.

### 3. Stable Source Identification Algorithm

This section presents a mathematical model for a circular conductive region made up of two homogeneous media and defines the forward and inverse problems for the problem (1)–(5).

#### 3.1. Mathematical Model

In this work, we consider a circular region $\Omega$ as a conductive medium composed of two homogeneous media, as illustrated in Figure 1. Specifically, we consider that $\Omega$ consists of two concentric circles centered at the origin with radii $R_1$ and $R_2$. $\Omega_1$ is the circle with radius $R_1$, and $\Omega_2$ corresponds to the difference between the circle with radius $R_2$ and the circle with radius $R_1$, i.e., $\Omega = \Omega_1 \cup S_1 \cup \Omega_2$, where $S_1$ is the interface that separates regions $\Omega_1$ and $\Omega_2$. The conductivities of regions $\Omega_1$ and $\Omega_2$ are denoted by $\sigma_1$ and $\sigma_2$, respectively.

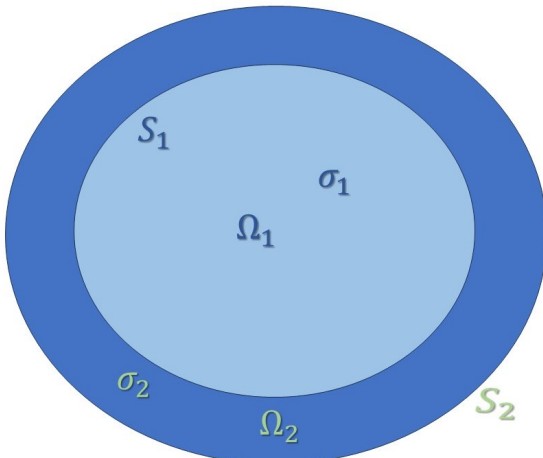

**Figure 1.** A circular conductive region composed of two homogeneous media. The source is located on the closed curve $S_1$, which separates the regions $\Omega_1$ and $\Omega_2$ with constant conductivities $\sigma_1$ and $\sigma_2$, respectively.

The boundary value problem to study the identification problem is given by [7]:

$$-\sigma_1 \Delta u_1 = 0, \quad \text{in} \quad \Omega_1, \tag{1}$$

$$-\sigma_2 \Delta u_2 = 0, \quad \text{in} \quad \Omega_2, \tag{2}$$

$$u_1 = u_2, \quad \text{on} \quad S_1, \tag{3}$$

$$\sigma_1 \frac{\partial u_1}{\partial n_1} = \sigma_2 \frac{\partial u_2}{\partial n_1} + g, \quad \text{on} \quad S_1, \tag{4}$$

$$\sigma_2 \frac{\partial u_2}{\partial n_2} = 0, \quad \text{on} \quad S_2, \tag{5}$$

where $g$ is the source, $u_i = u|_{\Omega_i}$, $i = 1, 2$, and $u$ represents the electric potential in $\Omega$. The symbol $\Delta$ represents the Laplace operator, which is also denoted as $\nabla^2$. Boundary condition (3) corresponds to the conductivity of the potential, and boundary condition (4) is associated with the jump in current flow due to the presence of the source. The conductivity of $\Omega^c$ is assumed to be zero, which leads to boundary condition (5). Using Green's formulas, we obtain the following compatibility condition:

$$\int_{S_1} g \, ds = 0. \tag{6}$$

Problem (1)–(5) is known as the superficial boundary value problem [7]. This problem has been used to study the inverse source problem in electroencephalography for cortical sources [2]. The following definitions are related to problem (1)–(5).

Given $g$ defined on $S_1$, the forward problem involves finding a measurement $V = u|_{S_2}$, where $u$ is the solution to problem (1)–(5).

Given a function $V$ defined on $S_2$, the inverse problem involves determining a source $g$ defined on $S_1$ such that the solution $u$ to the forward problem corresponding to $g$ satisfies that $u|_{S_2} = V$.

### 3.2. Forward Problem

In this section, we consider the following functional spaces:

$$L_2(S_i) = \left\{ h : S_1 \to \mathbb{R} : \langle h, h \rangle_{L_2(S_i)} = \int_{S_i} h^2(x)dx < \infty \right\}, i = 1, 2,$$

$$L_{2,\perp}(S_i) = \left\{ h \in L_2(S_i) : \langle h, 1 \rangle_{L_2(S_i)} = 0 \right\}, i = 1, 2,$$

$$L_2(\Omega) = \left\{ u : \Omega \to \mathbb{R} : \langle u, u \rangle_{L_2(\Omega)} = \int_\Omega u^2(x)dx < \infty \right\},$$

$$H^1(\Omega) = \{ u \in L_2(\Omega) : \text{its derivative is in } L_2(\Omega) \},$$

$$H^{1,\perp}(\Omega) = \left\{ u \in H^1(\Omega) : \langle u, 1 \rangle_{L_2(\Omega)} = 0 \right\},$$

where $\langle \cdot, \cdot \rangle$ is the inner product in the space indicated by the subscript.

To find the solution to problem (1)–(5), we consider that

$$g(\theta) = \sum_{k=1}^{\infty} g_k^1 \cos k\theta + g_k^2 \sin k\theta, \tag{7}$$

where $g_k^1$ and $g_k^2$ are the Fourier coefficients of $g$. The solution to the forward problem is given by

$$\begin{aligned} V(\theta) = A(g)(\theta) &= u(R_2, \theta) \\ &= \sum_{k=1}^{\infty} a_k g_k^1 \cos k\theta + a_k g_k^2 \sin k\theta, \end{aligned} \tag{8}$$

where

$$a_k = \frac{2R_1^{k+1} R_2^k}{k[(\sigma_1 - \sigma_2)R_1^{2k} + (\sigma_1 + \sigma_2)R_2^{2k}]}. \tag{9}$$

The linear operator $A : L_{2,\perp}(S_1) \longrightarrow L_{2,\perp}(S_2)$ is defined as $A(g) := u|_{S_2}$, where $u \in H^{1,\perp}(\Omega)$ is the solution to problem (1)–(5).

### 3.3. Stable Algorithm for the Inverse Source Problem

For the inverse problem, we consider that the exact (ideal) measurement $V(\theta) = \sum_{k=1}^{\infty} V_k^1 \cos k\theta + V_k^2 \sin k\theta$ is known, where $V_k^1$ and $V_k^2$ are the Fourier coefficients of $V$. Using Equation (8), we obtain the Fourier coefficients of the source, $g$.

$$\begin{aligned} g_k^i = \frac{V_k^i}{a_k} &= \frac{k(\sigma_1 - \sigma_2)}{2R_1} \left( \frac{R_1}{R_2} \right)^k V_k^i + \\ &\quad \frac{k(\sigma_1 + \sigma_2)}{2R_1} \left( \frac{R_2}{R_1} \right)^k V_k^i, \ i = 1, 2. \end{aligned} \tag{10}$$

In practice, the measurement has errors for various reasons, such as errors from the measurement device, truncation errors, and the application of filters to eliminate unwanted signals (signal contamination). The error in the measurement is reflected in its Fourier

coefficients, which must be considered carefully since the inverse source problem is numerically unstable. More specifically, the error in the coefficients is reflected in the term $\left(\frac{R_2}{R_1}\right)^k$, $k = 1, 2, ...$ of the coefficients in Equation (10). Note that as these terms grow with increasing $k$, even a small error in the measurement $V$ can result in significant changes to the location of the source. More precisely, if we know $V_\delta$ instead of $V$ and consider measurement $V_\delta(\theta) = \sum\limits_{k=1}^{\infty} V_{k,\delta}^1 \cos k\theta + V_{k,\delta}^2 \sin k\theta$, with $\|V - V_\delta\|_{L_2(S_2)} \leq \delta$, the Fourier coefficients of the recovered sources are given by Equation (11):

$$g_{k,\delta}^i = \frac{V_{k,\delta}^i}{a_k} = \frac{k(\sigma_1 - \sigma_2)}{2R_1}\left(\frac{R_1}{R_2}\right)^k V_{k,\delta}^i + \frac{k(\sigma_1 + \sigma_2)}{2R_1}\left(\frac{R_2}{R_1}\right)^k V_{k,\delta}^i, \quad i = 1, 2. \tag{11}$$

However, the terms $\left(\frac{R_2}{R_1}\right)^k V_{k,\delta}^i$ amplify the errors, and the series of the recovered source may not converge. To address this numerical instability, the Tikhonov functional is used [5]:

$$J_{\alpha(\delta)}(g) = \frac{1}{2}\|A(g) - V\|_{L_2(S_1)}^2 + \frac{\alpha(\delta)}{2}\|g\|_{L_2(S_2)}^2, \tag{12}$$

where $\alpha(\delta) > 0$ is the Tikhonov regularization parameter, which can be chosen using the L-curve criterion [22], and $\|\cdot\|_{L_2(S_i)}^2$ denotes the norm of $L_2(S_i)$, for $i = 1, 2$. To find the unique minimum of $J_{\alpha(\delta)}$, we must solve the normal equations [5] given by:

$$[A^*A + \alpha(\delta)I](g) = A^*V, \tag{13}$$

where $A^* : L_{2,\perp}(S_2) \to L_{2,\perp}(S_1)$ is the adjoint operator of $A$, which is given by [7]:

$$A^*(h)(\theta) = \frac{R_2}{R_1}\sum\limits_{k=1}^{\infty} a_k h_k^1 \cos k\theta + a_k h_k^2 \sin k\theta, \tag{14}$$

where $h(\theta) = \sum\limits_{k=1}^{\infty} h_k^1 \cos k\theta + h_k^2 \sin k\theta$, and $a_k$ is given by Equation (9).

After substituting into the normal Equations (13), we obtain the regularized solution:

$$g_{\alpha(\delta)}(\theta) = \sum\limits_{k=1}^{\infty} A_k(\alpha)\left[V_{k,\delta}^1 \cos k\theta + V_{k,\delta}^2 \sin k\theta\right], \tag{15}$$

where

$$A_k(\alpha) = \frac{a_k R_2}{(a_k)^2 R_2 + \alpha(\delta)R_1}. \tag{16}$$

**Remark 1.** *When $\alpha = 0$ in (15), the Fourier coefficients of the recovered source coincide with the coefficients given in Equation (11). Equation (15), obtained from the Tikhonov functional (12), provides us with a stable algorithm for recovering the regularized source $g_{\alpha(\delta)}$ from the measurement $V_\delta$. This algorithm will be implemented on the FPGA, and in the next section, we will illustrate algorithm (15) by implementing examples in MATLAB as a first step.*

## 4. Numerical Examples: MATLAB Implementation

In this section, we illustrate the source identification algorithm by considering the forward and inverse problems for different sources. We developed MATLAB programs to validate the algorithm for its computational implementation. The following section will describe the FPGA implementation of the source identification algorithm, and we will

compare the hardware implementation with the MATLAB implementation using the same examples. The region $\Omega$ is described in Section 3.1, and it is shown in Figure 1.

To illustrate the algorithm, we built synthetic examples as follows:

1. We took some values for parameters $\sigma_1$, $\sigma_2$, $R_1$, and $R_2$ and defined a source $g$ on $S_1$.
2. We solved the boundary value problem (1)–(5).
3. We computed the exact measurement $V = u|_{S_2}$ using Equation (8) for $N = 16$, which was chosen by numerical tests.
4. To emulate the measurement with error, we added an appropriate random error to the coefficients $V_k^1$ and $V_k^2$ (where $k = 1, 2 \ldots$) using the *rand* function of MATLAB. Hence, we obtained the coefficients $V_{k,\delta}^1$ and $V_{k,\delta}^2$, $k = 1, 2 \ldots$ of the measurement with error $V_\delta$, which satisfies $\|V_\delta - V\|_{L_2(S_2)} \leq \delta$.
5. We obtained the regularized solution to the inverse problem by taking $N = 16$ in Equation (15), i.e., we used

$$g_{\alpha(\delta),N}(\theta) = \sum_{k=1}^{16} A_k(\alpha) \left[ V_{k,\delta}^1 \cos k\theta + V_{k,\delta}^2 \sin k\theta \right], \tag{17}$$

as an approximate (regularized) solution, where $A_k(\alpha)$ is given in (16). In the examples below, we considered $\delta = 0.1$ and $\alpha = 10^{-3}$ (which were chosen numerically).

In the plots presented in the following examples, the magnitudes are either dimensionless or have a dimension of one. In other words, they are quantities that do not have an associated physical dimension. Consequently, they are pure numbers that can describe a physical characteristic without a dimension or an explicit unit of expression.

**Example 1.** In this example, we set $R_1 = 1$, $R_2 = 1.2$, $\sigma_1 = 3$, and $\sigma_2 = 1$ and considered the exact source $g$ given by

$$g(x,y) = f(x,y) - \frac{1}{m(S_1)} \int_{S_1} f(x,y)ds, \tag{18}$$

for all $(x,y) \in S_1$, where $f(x,y) = e^{-\frac{\|(x,y) - (a_1,a_2)\|^2}{2\beta^2}}$ for $(x,y) \in S_1$, $(a_1, a_2) = (0,1) \in S_1$, $\beta^2 = 0.1$, and $m(S_1) = 2\pi R_1$. In polar coordinates $(r,\theta)$, $g$ is given by

$$g(\theta) = f(R_1,\theta) - \frac{1}{2\pi} \int_0^{2\pi} f(R_1,\theta)d\theta, \text{ for all } \theta \in [0, 2\pi], \tag{19}$$

where $f(R_1,\theta) = e^{\frac{R_1^2 \cos(\theta - \theta_0)}{\beta^2}}$ for all $\theta \in [0, 2\pi]$, $\theta_0 = \frac{\pi}{2}$, and $\beta^2 = 0.1$.

We approximated the exact source $g$ by its truncated Fourier series $g_N$ using the first $N = 16$ terms. The Fourier coefficients $g_k^1$, $g_k^2$, $k = 1, 2, \ldots, N$ were obtained numerically using the *quadl* function of MATLAB. In this case, the exact measurement $V$ was generated using Equation (17).

The relative error between the exact source $g$ and the recovered source $g_{\alpha(\delta),N}$, denoted by $RE_{S_1}(g_{\alpha(\delta),N}, g)$, is given by

$$RE_{S_1}(g_{\alpha(\delta),N}, g) = \|g_{\alpha(\delta),N} - g\|_{L_2(S_1)} / \|g\|_{L_2(S_1)}$$
$$= 0.12.$$

Figure 2 displays both the exact measurement $V$ and the measurement with error $V_\delta$, where $\delta = 0.1$.

Figure 3 displays the plot of both the exact source and the recovered source without regularization. We observe the necessity of applying regularization methods.

Figure 4 displays the plot of both the exact and the recovered source with regularization.

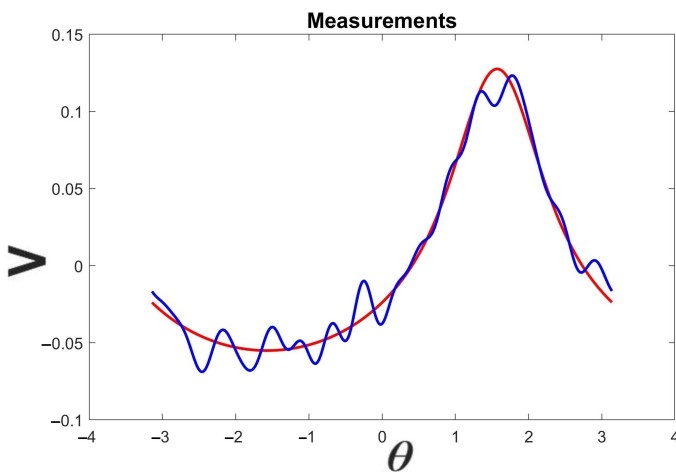

**Figure 2.** Exact measurement is shown in red, and recovered measurement is in blue. The error in the exact measurement was obtained by adding a random error to the exact measurement using the rand function of MATLAB.

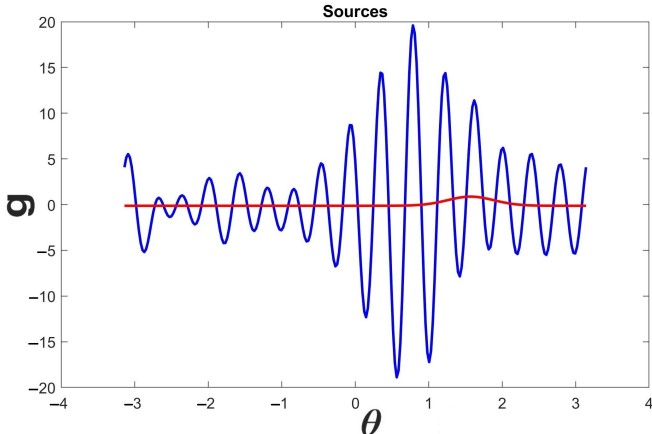

**Figure 3.** Exact source is shown in red, and recovered source without regularization is in blue. In this case, the recovered source was obtained using the coefficients given by Equation (11), and it was far from the exact source due to the numerical instability of the inverse source problem.

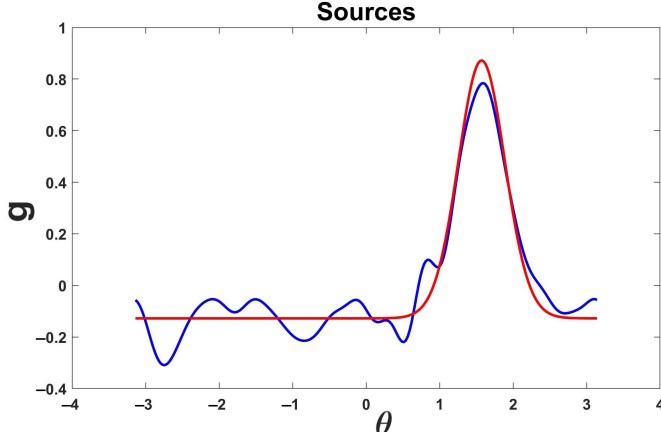

**Figure 4.** Exact source is shown in red, and recovered source with regularization is in blue. In this case, the regularized source given by (17), which was obtained using Tikhonov regularization, allowed us to handle numerical instability to get a good approximation of the exact source.

**Example 2.** We considered the same values for $R_1$, $R_2$, $\sigma_1$, and $\sigma_2$ as in the previous example. We considered the following function: $g(x, y) = ye^x + e^y + x^2$, which in polar coordinates is

given by $g(\theta) = sin(\theta)e^{cos(\theta)} + e^{sin(\theta)} + cos^2(\theta)$. Using Equation (8), we found the solution to the forward problem $V$, i.e., to find the ideal (exact) measurement $V$, we had to find the solution to problem (1)–(5) and then restrict it to the boundary $S_2$. The noisy data $V_\delta$ was obtained as in the previous example. Figure 5 displays both the exact measurement $V$ (in red) and the measurement with error $V_\delta$ (in blue).

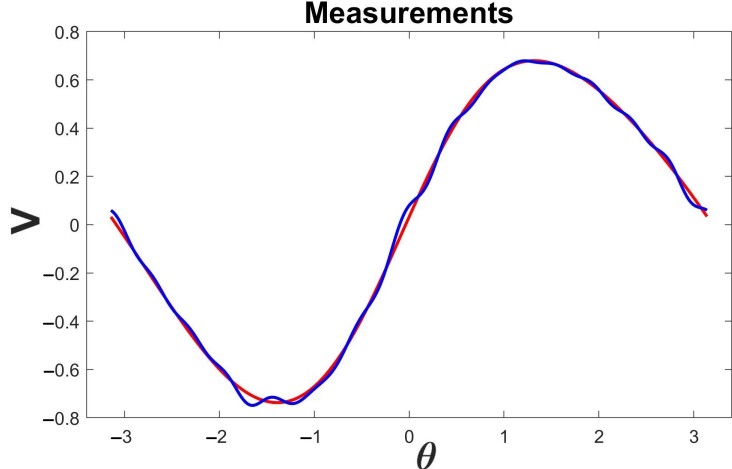

**Figure 5.** Exact measurement is shown in red, and measurement with error is in blue. The latter was obtained by adding a random error to the exact measurement, which was done using the rand function of MATLAB.

Figure 6 shows the plot of both the exact source and the recovered source without regularization. As in the previous example, we also observed the necessity of applying regularization methods.

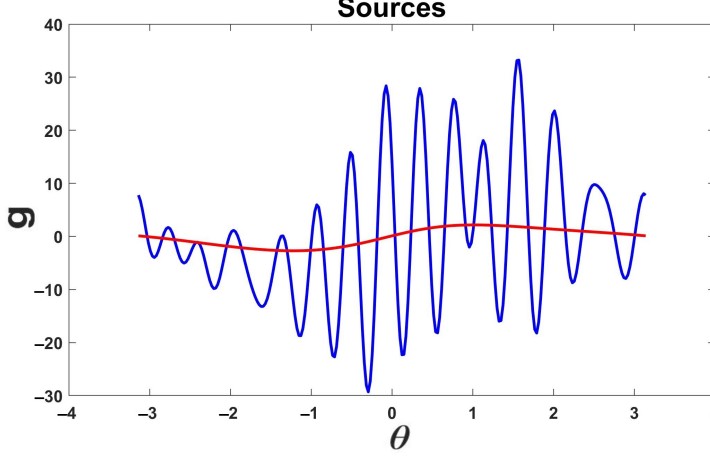

**Figure 6.** Exact source is shown in red, and recovered source without regularization is in blue. The difference shows the necessity to apply regularization methods.

Figure 7 displays both the exact and recovered sources when regularization was applied. The relative error between the exact source $g$ and the recovered regularized source, $g_{\alpha(\delta),N}$, denoted by $RE_{S_1}(g_{\alpha(\delta),N}, g)$, is given by

$$RE_{S_1}(g_{\alpha(\delta),N}, g) = 0.054.$$

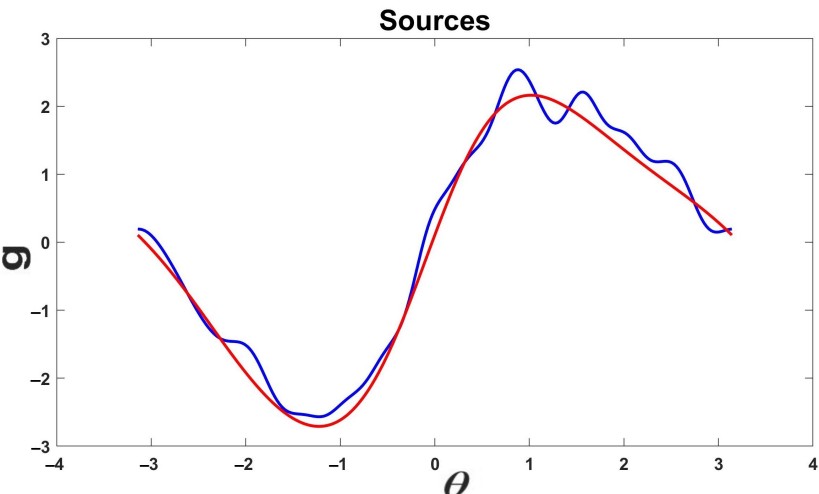

**Figure 7.** Exact source is shown in red, and the recovered source with regularization is in blue. In this case, the regularized source given by Equation (17) allowed us to obtain a stable approximation of the source.

**Example 3.** We considered the same values for $R_1$, $R_2$, $\sigma_1$, and $\sigma_2$ as in the previous example. We considered the following square-wave function given in polar coordinates as

$$g(\theta) = \begin{cases} -1, & \text{if } -\pi \leq \theta < 0, \\ 1, & \text{if } \quad 0 \leq \theta < \pi. \end{cases} \tag{20}$$

We found the solution $V$ to the forward problem and the noisy data $V_\delta$ for the boundary $S_2$ as in the previous examples. Figure 8 displays both the exact source $g$ (red) and its approximation by Fourier series $g_N$ (blue) when truncating the Fourier series (7) to the first N = 16 terms. Figure 9 shows both the exact measurement $V$ (in red) and the measurement with error $V_\delta$ (in blue).

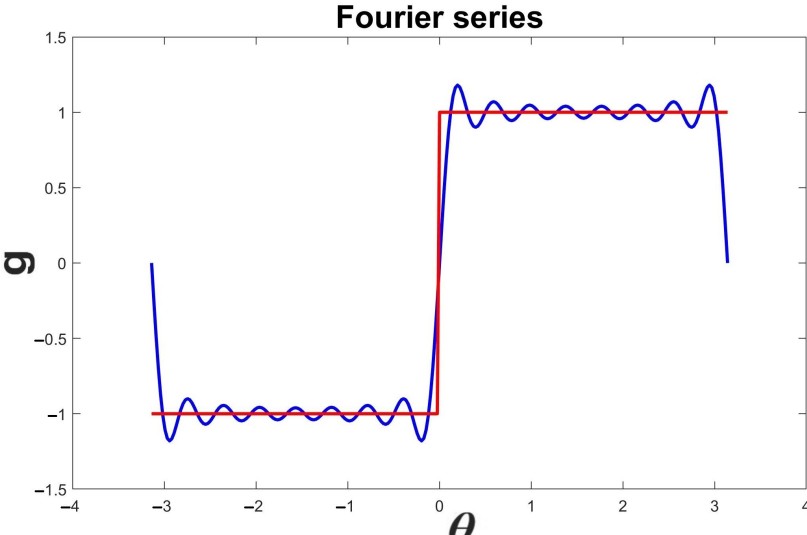

**Figure 8.** Plot of the exact square-wave function $g$ (red) and its approximation by Fourier series (blue). We observe the Fourier series oscillations and the presence of the Gibbs phenomenon. This jump function is frequently used in electronic signals.

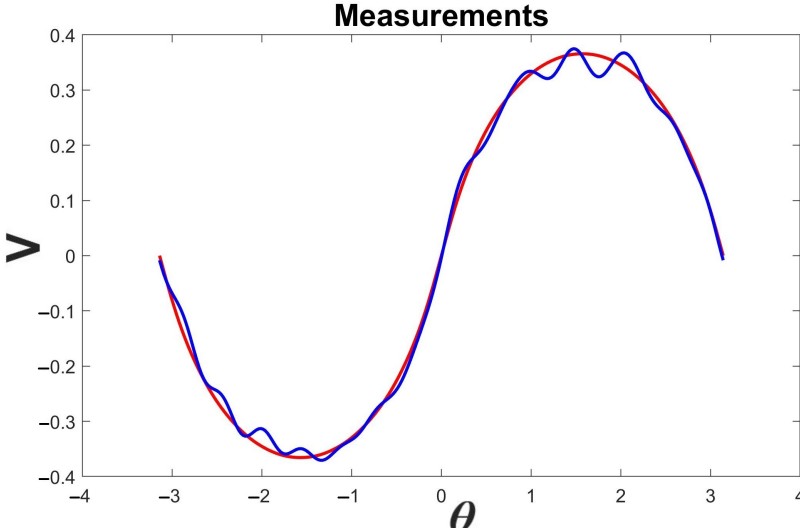

**Figure 9.** Exact measurement is shown in red, and measurement with error is in blue. The measurement with error was obtained by adding a random error to the exact measurement, which was done using the MATLAB function rand.

Figure 10 displays both the exact and recovered sources without regularization. As in the previous examples, we observed the importance of regularization methods to obtain stable solutions to the inverse problem.

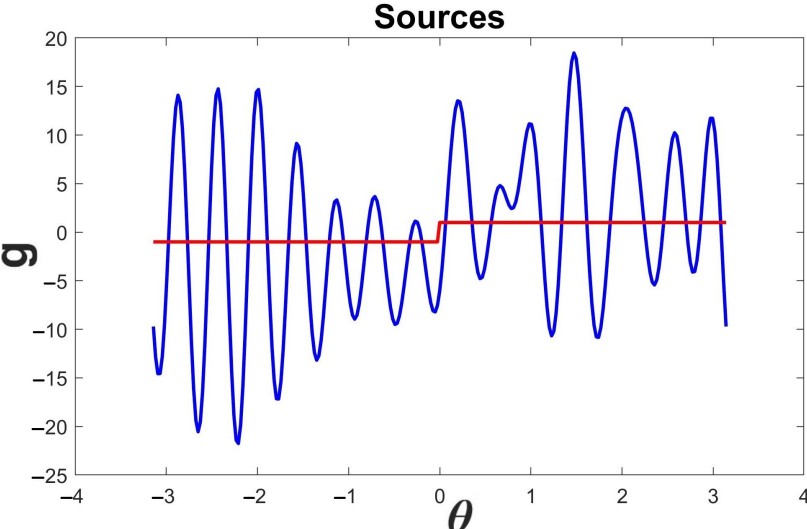

**Figure 10.** Implementation in FPGA: exact source is shown in red, and recovered source without regularization is in blue. The results obtained are similar to those found using MATLAB 2013a, which are shown in Figure 3.

Figure 11 displays both the exact and recovered sources when regularization was applied. The relative error between exact source $g$ and recovered source $g_{\alpha(\delta),N}$, denoted by $RE_{S_1}(g_{\alpha(\delta),N}, g)$, is given by

$$RE_{S_1}(g_{\alpha(\delta),N}, g) = 0.051.$$

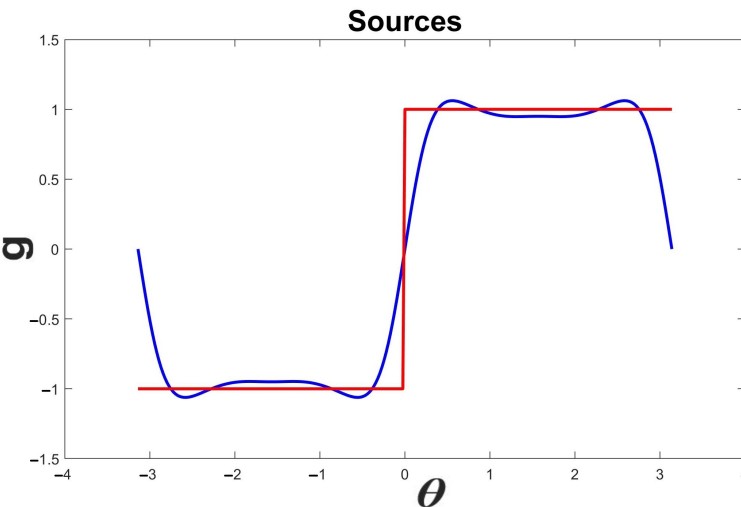

**Figure 11.** Exact source is shown in red, and the recovered source with regularization is in blue. The Tikhonov regularization method was used to handle numerical instability and to obtain a good approximation of the exact source. The regularized solution acted as a smoothing filter.

### 5. FPGA Implementation

Implementing algorithms on hardware devices is crucial as it optimizes the use of hardware resources. For instance, we can determine the number of coefficients in a series to achieve an appropriate approximation of and the number of bits used to represent a real number (precision). We can also ascertain if the implementation can be duplicated for multichannel problems.

Analyzing the arithmetic operations in Equation (17), we conclude that a feedback implementation is appropriate, as it minimizes hardware resources. The operations in Equation (17) will be repeated as follows:

1. Perform the products $A_1 V_1^1 cos(\theta)$ and $A_1 V_1^2 sin(\theta)$. The two's complement format is chosen for number representation, as the algorithm involves signed arithmetic operations.
2. Sum the results from the previous step.
3. Temporarily store the result.
4. Perform the products $A_2 V_2^1 cos(2\theta)$ and $A_1 V_2^2 sin(2\theta)$ and add them to the temporary result.
5. Repeat the process until term $N = 16$.

The advantage of this implementation is that the arithmetic modules can be reused. The hardware implementation components include:

1. A double-memory ROM block to store the coefficients $A_k V_k^1$ and $A_k V_k^2$.
2. A Xilinx direct digital synthesizer (DDS) module (see [23]) to generate the values of $sin(k\theta)$ and $cos(k\theta)$ for $k = 1, \dots, 16$.
3. Multiplexers to maintain synchrony in the control section.

Considering this, Figures 12 and 13 show the required hardware resources for architecture Modes 1 and 2, respectively.

The implementation results presented were obtained using the VHDL (Very High Speed Integrated Circuit Hardware Description Language) code synthesized on the Vivado IDE c2018.2.2 (64 bit) and tested on the Xilinx® 7 FPGA series. This series comprises four FPGA families that address a complete range of system requirements ranging from low-cost, small form factor, cost-sensitive, high-volume applications to ultra-high-end connectivity bandwidth, logic capacity, and signal processing capability for the most demanding high-performance applications: Spartan-7 xc7s100fgga484, Virtex-7 xc7v585tffg1157, Kintex-7 xc7k70tfbg484, and Artix-7 xc7a35tcpg236. Tests were carried out using a 100 MHz base clock.

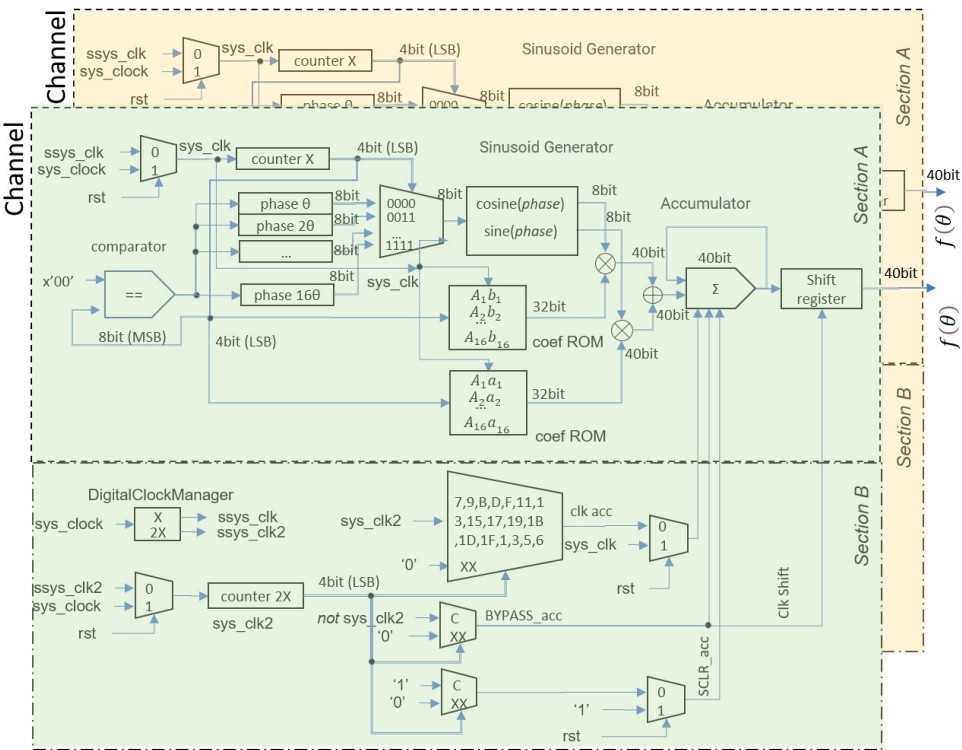

**Figure 12.** Feedback architecture, Mode 1: In this mode, the control unit (Section B), as well as the phase generation and the processing unit, are independent in each channel. This allows for operation at different speeds.

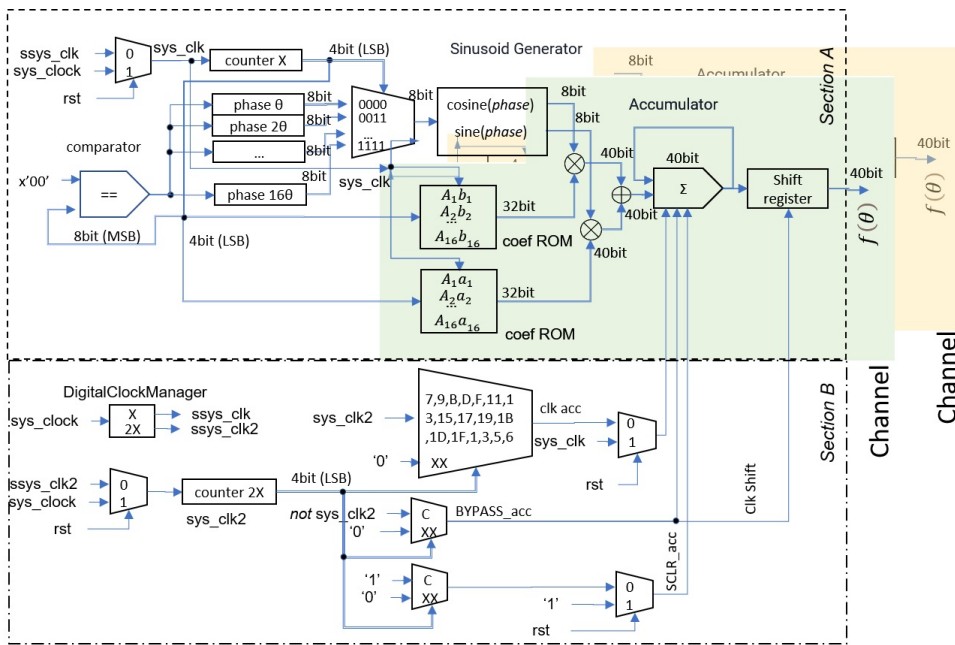

**Figure 13.** Feedback architecture, Mode 2: In this mode, the control section is shared with the algorithm operations section, resulting in coupled synchronization of the trigonometric base operations.

The instrumentation developed for the implementation of the algorithm consists of 48 coefficients involved in expression (13) with $N = 16$. Each coefficient is 16 bits long. Therefore, there are 16 coefficients for $A_k$, $V_k^1$, and $V_k^2$. The DDS component was configured with a partition of 8 bits for the phase and amplitude of the sinusoidal functions.

The product requires 40 bits since we have 32-bit and 8-bit numbers. Finally, both the accumulated sum and the resulting output require 40 bits.

*5.1. Architecture Description*

Figures 12 and 13 show the architectures for Modes 1 and 2, respectively. Both architectures contain two channels, with each composed of:

1.    Section A: This section contains the arithmetic operations.
2.    Section B: This section controls the operations in Section A to synchronize the pipeline operations.

The design of the architecture consists of three blocks:

1.    The trigonometric base.
2.    Linear combination of the elements of the base.
3.    Control module.

Trigonometric base: The counterX module performs the following three fundamental actions through its four least-significant bits:

1.    Acts as an 8-bit selector control in the 16-to-1 multiplexer.
2.    Increments the consecutive value from 1 to 16 for the resolution.
3.    Synchronizes the addressing and reading of the ROM memory.

To evaluate the functions $cos(\theta)$ and $sin(\theta)$ and address the ROM memory, a latency of two cycles is used. This maintains synchronization between the DDS module and the ROM memories.

Linear combination of the trigonometric base: The following latencies are required for different operations:

1.    A two-cycle latency for the DDS module to reflect the sine and cosine values on the data bus and for the ROM memory module to reflect the coefficients simultaneously on the data bus. Both modules work in parallel.
2.    A one-cycle latency for each operation (product and sum). These operations are performed in series.
3.    A one-cycle latency to store the result in the accumulator.
4.    A latency of 16 cycles to obtain $g(\theta)$.

Thus, a new value of $g(\theta)$ is calculated every 16 cycles. With the main clock of the used FPGA operating at 100 MHz, each point in the partition has an output frequency of 6.25 MHz.

Construction of the control section: The control section, shown in Section B of Figures 12 and 13, corresponds to the synchronization of the algorithm. Its design allows for efficient use of throughput, which provides the amount of data processed in each clock cycle. The pipeline plays a crucial role in this section, as input phase-generation data must be synchronized to get output data, which correspond to new values of $g(\theta)$. Output data are generated in this form, which must be aligned. The processing stages of the control section must be applied in each clock cycle.

The control section is utilized to manipulate phase shifts of the main clock to ensure overall synchronization of the architecture. The design operates using two clock signals generated by the digital clock manager block. The signal *sys_clk* is used for constructing both the *trigonometric base* and the *linear combinations of the trigonometric base* sections. To maintain synchronizations of the accumulator and shift register elements, *ssys_clk*2 must be twice as fast as *ssys_clk*. The multiplexer array *ssys_clk*2 serves as a control element for resetting the accumulator content, clock signal, and bypass signal. The bypass signal allows for passing on sum results to shift the register, which results in obtaining the first phase sum.

We developed two architectures, labeled Mode 1 and Mode 2, which are identical in the case of a single channel. For multiple channels, the Mode 2 architecture takes advantage of the characteristics of the algorithm to reduce hardware consumption. This reduction

can be observed in Section A of Figure 13, where the generation of the functions $\sin kx$ and $\cos kx$, where $k = 1, 2, \ldots$, is shared across different channels. The control module, shown in Section B, is shared by both architectures. The Mode 1 architecture allows for operation at different speeds, which is beneficial when working with either fast or slow signals. The Mode 2 architecture reduces resource consumption by exploiting the characteristics of the source identification algorithm.

### 5.2. Resource Description

The resources of the Xilinx 7 series FPGAs are:

1.  LUTs (lookup tables): These contain the logical elements that determine the output from one or multiple inputs. They are essentially truth tables created from the description of the VHDL program.
2.  FFs (flip flops): Sequential logical elements with one bit of memory.
3.  RAM blocks (random access memory): Each block has a storage capacity of 32 *K*-bit.
4.  DSP blocks (digital signal processing): Specialized blocks for product, sum, and accumulation operations for signed numbers in two's complement format. These operations are called: multiply–accumulate (MAC).
5.  Power consumption: Determines the energy consumption of the system.

From Section B in Figure 12, it can be seen that the channel blocks share control, which synchronizes MAC operations, phase generation, and sine and cosine values, i.e., the trigonometric functions that make up the base. Combining the common control operations with sine and cosine generation reduces the number of hardware resources.

Figures 14 and 15 show the resources used in the proposed architectures. In the case of one channel, both architectures have identical resource consumption. Architecture Mode 2 presents lower resource consumption for more than one channel.

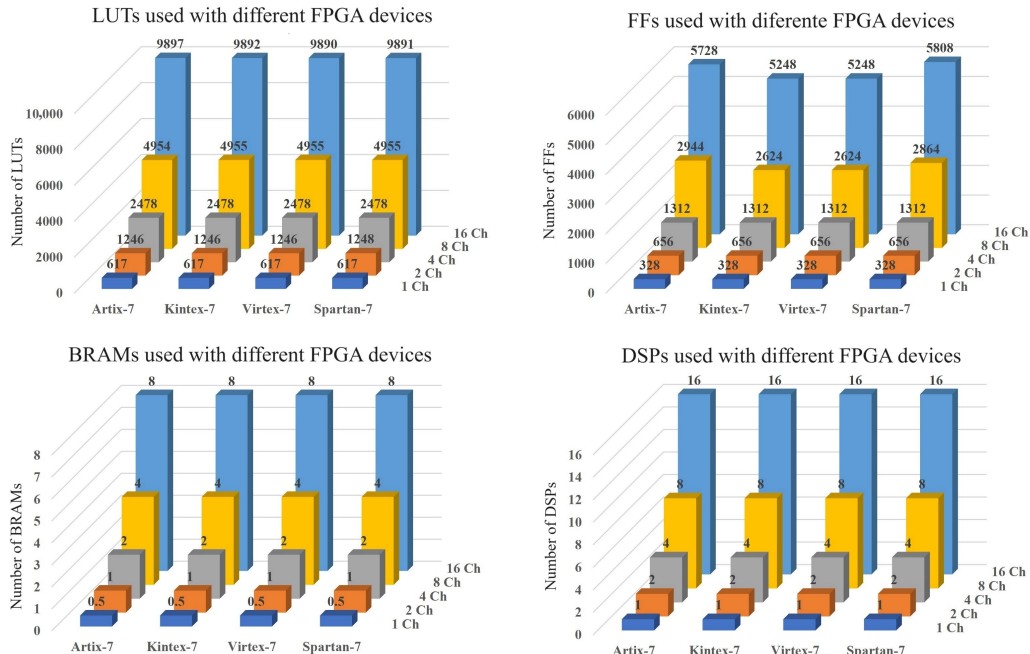

**Figure 14.** Resources, Mode 1: We observe that as the number of channels increases, resource consumption linearly increases in all FPGAs, and there is no difference in consumption.

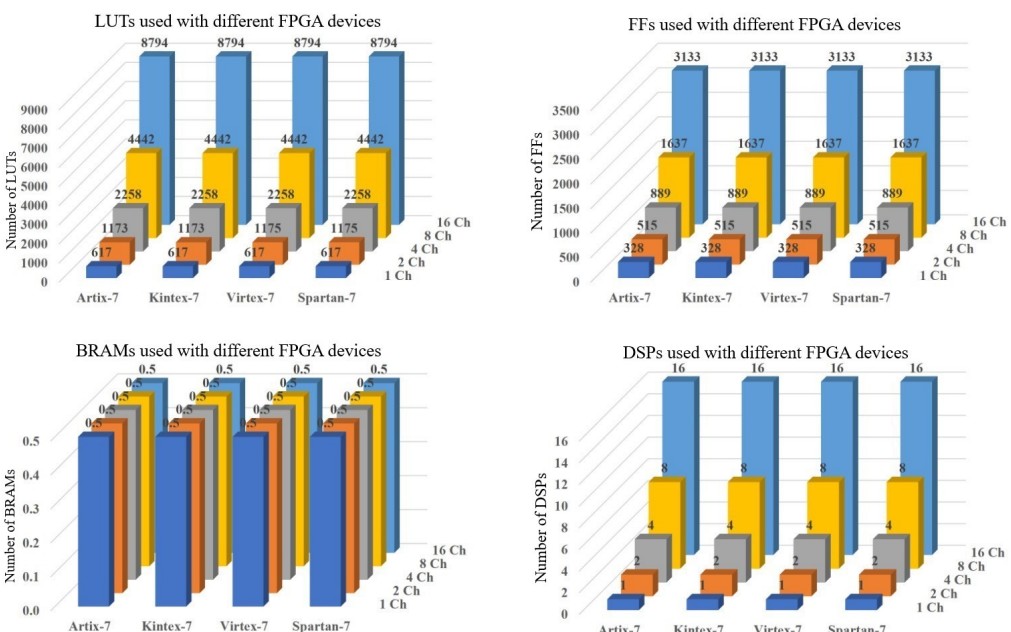

**Figure 15.** Resources, Mode 2: We observe that resource consumption is lower for LUTs, FFs, and BRAM (block RAM) compared to Mode 1 as the number of channels increases. In particular, there is no increase in BRAM consumption as the architecture uses the same resources. The DSP consumption is identical for both architectures.

Figure 16 shows the results of the following relation:

$$\frac{\text{mode } 1 - \text{mode } 2}{\text{mode } 1} = 1 - \frac{\text{mode } 2}{\text{mode } 1}. \tag{21}$$

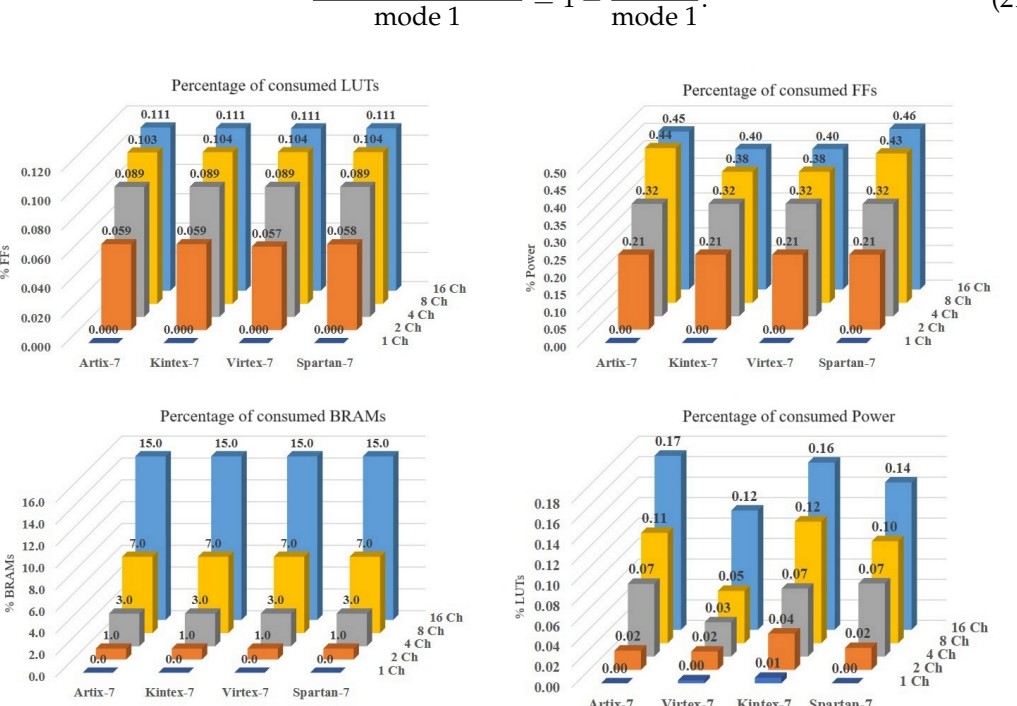

**Figure 16.** Using Equation (21), we calculated the percentage of resources consumed by Mode 2 compared to Mode 1. As the number of channels increases, this percentage also increases. For LUTs, the percentage is nearly identical. For FFs, both Kintex-7 and Virtex-7 have identical performance.

The relation gives the percentage of resources that Mode 2 consumes compared to Mode 1. For two channels, 5.9% fewer LUT resources are obtained for Artix and Kintex. For four channels, 8.9% fewer LUT resources are obtained for all FPGAs. For 16 channels, 11% fewer LUT resources are obtained for all FPGAs.

For two channels, 21% fewer FF resources are obtained for all FPGAs. For 16 channels, 40% to 46% fewer FF resources are obtained. For 16 channels, BRAM consumption is reduced by a factor of 15, as architecture Mode 2 uses only one DDS module. For 16 channels, power consumption is reduced by 17%, 12%, 16%, and 14% in Artix, Virtex, Kintex, and Spartan, respectively.

Figure 17 shows the power consumption. Mode 2 has lower current consumption. For both modes, Artix-7 has the lowest consumption.

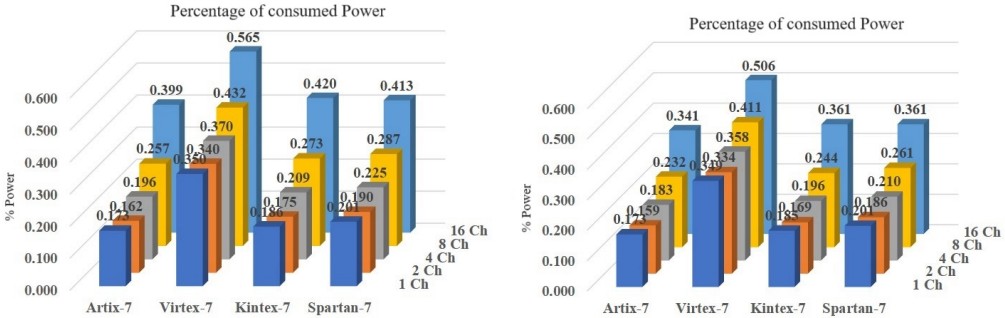

**Figure 17.** Left: Power, Mode 1. Right: Power, Mode 2. We observe that Mode 2 has lower current consumption. For both modes, Artix-7 has the lowest consumption.

## 6. Validation of the Hardware Implementation

In this section, we present the results of the FPGA implementation of the same examples given in Section 4. We validate those examples by using MATLAB.

Figures 18–22 show the examples from Section 4. Figures 23–25 show the error between the recovered sources using MATLAB and the FPGA.

Figures 19, 21, and 26 show the recovered sources without regularization provided by the FPGA. We can see that the recovered sources are far from the exact source from the MATLAB implementation. Figures 18, 20, and 22 show the recovered sources provided by the FPGA implementation using regularization. We emphasize that we obtained similar results from MATLAB.

We define the norm of maximum absolute error (NMAE) between the regularized recovered source $g_{\alpha,N}$ and the recovered implemented source $g_{\alpha,N}^I$ as follows:

$$\text{NMAE} = \max_{\theta}\left\{\left|g_{\alpha,N}(\theta) - g_{\alpha,N}^I(\theta)\right|\right\}.$$

We define the maximum absolute error (MAE) as:

$$\text{MAE}(\theta) = \left|g_{\alpha,N}(\theta) - g_{\alpha,N}^I(\theta)\right|, \text{for all } \theta \in [-\pi, \pi].$$

Figures 23–25 show the MAE between the sources recovered using MATLAB and the sources recovered using the FPGA implementation. We observe that the NMAE is less than $10^{-3}$. This result confirms that the FPGA implementation of the algorithm is reliable.

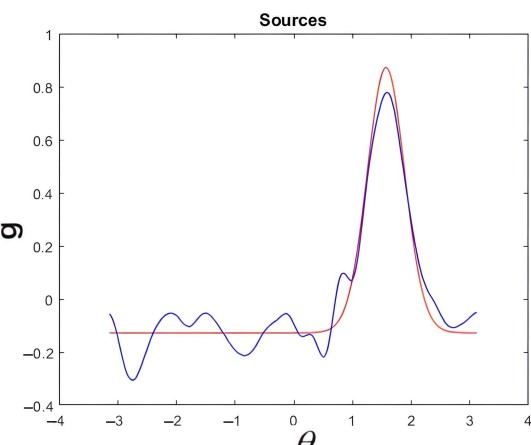

**Figure 18.** The exact source is shown in red, and the result provided by the FPGA implementation using Equation (17) with regularization is shown in blue. Results similar to those from the MATLAB implementation are obtained, thus validating the FPGA implementation.

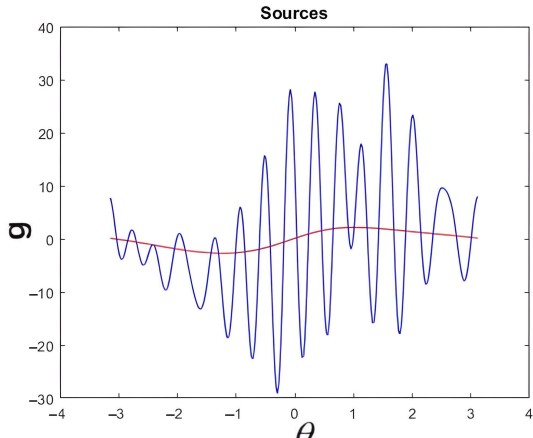

**Figure 19.** Exact source is shown in red, whereas the blue line shows the approximate source obtained by the implementation in the FPGA using Equation (11) (without regularization). Results similar to those from the MATLAB implementation are obtained. Thus, we validate the FPGA implementation. The recovered source is obtained by applying the algorithm to the measured data.

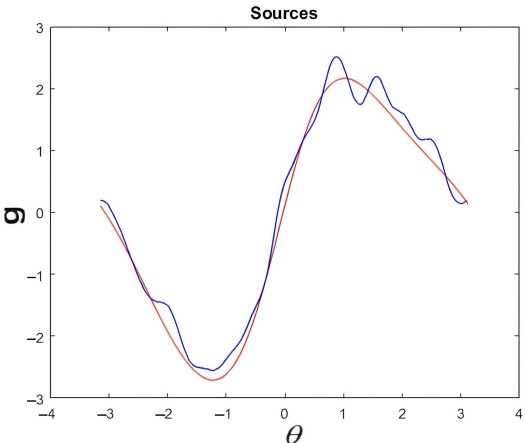

**Figure 20.** Exact source is shown in red. Source recovered by the FPGA implementation using Equation (17) with regularization is shown in blue. Results similar to those from the MATLAB implementation are obtained. Thus, we validate the FPGA implementation.

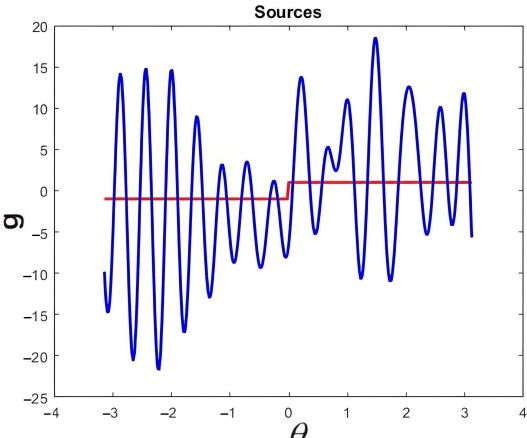

**Figure 21.** Exact source is shown in red, whereas the blue line shows the approximate source obtained by the FPGA implementation using Equation (11) without regularization. Results similar to those from the MATLAB implementation are obtained. Thus, we validate the FPGA implementation.

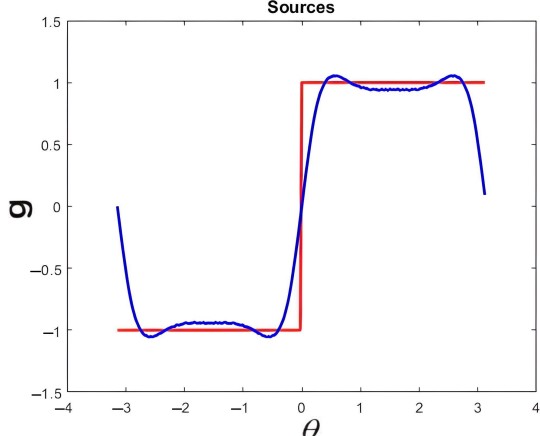

**Figure 22.** Exact source is shown in red, and result provided by the FPGA implementation using Equation (17) with regularization is in blue. Results similar to those from the MATLAB implementation are obtained. Thus, we validate the FPGA implementation.

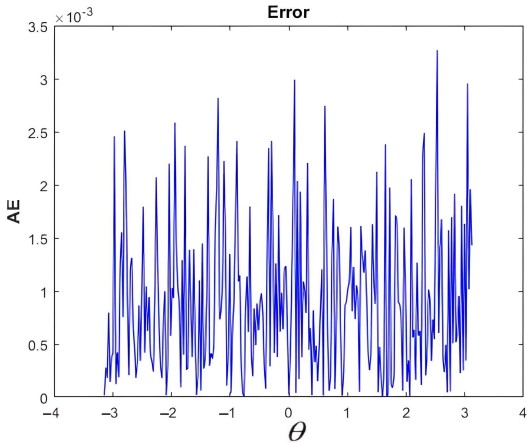

**Figure 23.** MAE between the source recovered using MATLAB programs and the one recovered by the FPGA implementation (note the scale on the y-axis). The NMAE is less than $10^{-3}$, which shows that the FPGA implementation gives almost the same results as the MATLAB implementation.

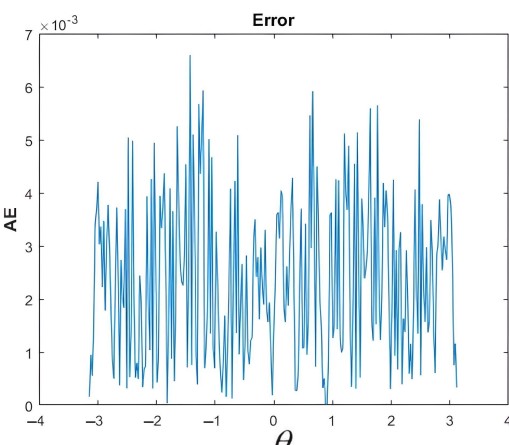

**Figure 24.** MAE between the source recovered using MATLAB programs and the one recovered by the FPGA implementation. Note the scale. The NMAE is less than $10^{-3}$, which shows that the FPGA implementation gives almost the same results as the MATLAB implementation.

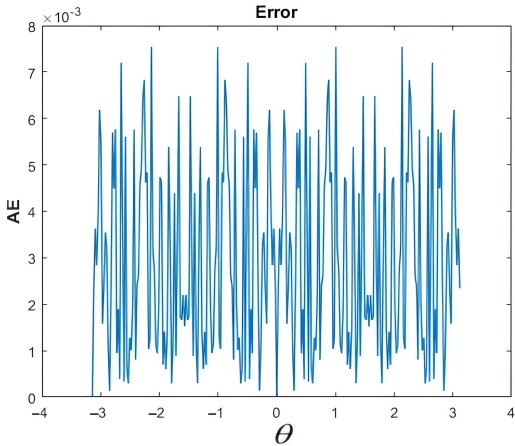

**Figure 25.** MAE between the source recovered using MATLAB programs and the one recovered by the FPGA implementation. Note the scale. The NMAE is less than $10^{-3}$, which shows that the FPGA implementation provides results that are comparable to the MATLAB implementation.

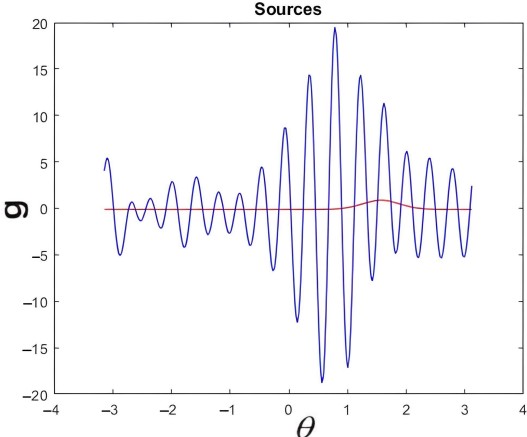

**Figure 26.** Exact source is shown in red, and the approximate source obtained by the FPGA implementation using Equation (11) without regularization is in blue. We obtained similar results to those of the MATLAB implementation, thus validating the FPGA implementation.

## 7. Discussion

The source identification problem arises in many applications, such as inverse electroencephalography, inverse electrocardiography, and inverse geophysics. An FPGA is an integrated circuit that allows the implementation of algorithms and other complex mathematical operations. In this work, we implemented a source identification algorithm considering a non-homogeneous medium made up of two homogeneous media and with sources located on the interface separating the homogeneous media. The algorithm is expressed as a Fourier series expansion that considers the circular geometry.

Since the source identification problem presents numerical instability, we used the Tikhonov regularization method and a cut-off of the Fourier expansion to handle the numerical instability. The Tikhonov regularization parameter and the truncation term were chosen through numerical testing.

This work presents a hardware architecture with two designs to implement the source identification algorithm. These architectures were tested on four FPGAs from the 7 Series of Xilinx: Spartan-7 xc7s100fgga484, Virtex-7 xc7v585tffg1157, Kintex-7 xc7k70tfbg484, and Artix-7 xc7a35tcpg236. Each architecture contains two channels, each of which is composed of two sections: one for the arithmetic operations and one for the control of the operations. We report the resource consumption of the 7 Series of Xilinx FPGAs. These resources include LUTs, RAM blocks, FFs, DSP blocks, and power consumption. One of the developed architectures can be used for problems with a larger number of channels. One example of this case is inverse electroencephalography, for which many inverse source problems must be solved per second.

To validate the hardware implementation, we constructed three synthetic examples in MATLAB. The results of both the hardware and software implementations were then compared in terms of error. These synthetic examples correspond to three types of functions that encompass a wide range of sources.

The first example is a trigonometric polynomial, while the second is a smooth function with rapidly decaying Fourier coefficients. The third example is a jump function, which represents a different class of functions. In this last case, the Fourier series exhibits the Gibbs phenomenon at the jump point. However, the regularization technique applied here acts as a smoothing mechanism, eliminating the Gibbs phenomenon in the recovered source.

As mentioned in the article, we conducted additional software and hardware implementations using similar functions and observed comparable results.

Given that we are considering two regularization parameters, we performed numerical tests to identify the pair that most effectively recovers the sources. We found that when the Tikhonov regularization parameter is fixed and the truncation parameter increases, the error between the exact source and the recovered source also increases. Similarly, when the truncation parameter is fixed, the results obtained for other regularization parameters are less accurate. Moreover, the precision of the hardware implementation does not improve when considering other pairs of parameters close to the chosen pair.

## 8. Conclusions

In this paper, we proposed two loop unrolling architectures for implementing a stable source identification algorithm on FPGAs. Our extensive literature review revealed no instances for which loop unrolling was utilized in the implementation of inverse source identification algorithms.

FPGA devices provide several advantages for algorithm implementation: the most significant being their ability to accelerate these algorithms. Loop unrolling architectures serve as effective strategies for task distribution to configurable hardware and thereby reduce overhead and facilitate parallel processing.

The first architecture (Mode 1) is an expanded implementation that encompasses all operations involved in the algorithm. In contrast, the second architecture (Mode 2) lever-

ages the algorithm's properties to apply a pipeline system and reuse hardware resources, thereby enhancing performance.

We tested the implementation using synthetic examples and obtained comparable numerical results for both architectures. The error between the exact source and the recovered source was found to be less than $10^{-3}$, which is deemed an acceptable result. One of the tested examples involved a square source function, for which the recovered source exhibited smoothing properties, as evidenced by the lack of variation in the Fourier series expansion.

A key advantage of the architectures developed in this study is our ability to determine the required number of coefficients, as this number significantly impacts the numerical stability of the algorithm. Furthermore, we can select the number of bits to achieve accurate approximations while minimizing hardware consumption. It is important to note that these architectures are designed to serve as a foundation for other algorithms. We infer that the architectures developed in this study can be repurposed for other similar algorithms arising from inverse source problems by leveraging their reconfigurable features.

The results presented here can be extended to concentric spheres and complex geometries. In these cases, the trigonometric base must be changed and implemented. For spheres, the base is the spherical harmonics, and for complex geometries, we can choose a base generated for the finite element method or the finite difference method. The second architecture (Mode 2) employs a control system based on multiplexors (as opposed to finite state machines), which reduces flip-flop resource consumption and design complexity and can be used for multichannel problems. Therefore, the architectures developed for implementing the inverse source algorithm can enable the creation of portable devices for problems in various fields such as engineering and medicine, as they can simplify the design and implementation process due to their reconfigurability.

**Author Contributions:** Conceptualization, J.J.O.-O., C.A.H.-G., M.M.M.-C., J.R.C.-S. and J.J.C.-M.; methodology, J.J.O.-O., C.A.H.-G., J.R.C.-S. and J.J.C.-M.; software, J.J.O.-O., J.R.C.-S. and J.J.C.-M.; validation, J.J.O.-O., J.R.C.-S. and J.J.C.-M.; formal analysis, J.J.O.-O., J.R.C.-S., C.A.H.-G., M.M.M.-C. and J.J.C.-M.; investigation, J.J.O.-O., J.R.C.-S., C.A.H.-G., M.M.M.-C. and J.J.C.-M.; resources, J.J.O.-O., J.R.C.-S., C.A.H.-G., M.M.M.-C. and J.J.C.-M.; data curation, J.J.O.-O., J.R.C.-S. and J.J.C.-M.; writing—original draft preparation, J.J.O.-O., J.R.C.-S., C.A.H.-G., M.M.M.-C. and J.J.C.-M.; writing—review and editing, J.J.O.-O., J.R.C.-S., C.A.H.-G., M.M.M.-C. and J.J.C.-M.; visualization, J.J.O.-O., J.R.C.-S., C.A.H.-G., M.M.M.-C. and J.J.C.-M.; supervision, J.J.O.-O., C.A.H.-G., J.R.C.-S. and J.J.C.-M.; project administration, J.J.O.-O., J.R.C.-S., C.A.H.-G., M.M.M.-C. and J.J.C.-M.; funding acquisition, J.J.O.-O., J.R.C.-S., C.A.H.-G., M.M.M.-C. and J.J.C.-M. All authors have read and agreed to the published version of the manuscript.

**Funding:** This The research was funded by the National Council of Science and Technology in Mexico (CONAHCYT) and Project 00221 VIEP-BUAP.

**Institutional Review Board Statement:** Not applicable.

**Informed Consent Statement:** Not applicable.

**Data Availability Statement:** The original contributions presented in the study are included in the article, further inquiries can be directed to the corresponding author.

**Conflicts of Interest:** The authors declare no conflicts of interest.

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
