# Peer review of "FPGA-Based Hardware Implementation of a Stable Inverse Source Problem Algorithm in a Non-Homogeneous Circular Region"

_applsci, doi:10.3390/app14041388_

Round 1

Reviewer 1 Report

Comments and Suggestions for Authors

This paper presented a stable inverse source problem method.  The paperis devoted to very interesting topic and contains some useful results. The method targets FPGA-based architecture which is a leading trend in this decade. Unfortunately, the paper is not free from some drawbacks. Some of the concerns are shown as follows to further refine the paper.

(1) The abstract should be concise and simply describe the research methodology, results, and contributions. Please revise it. I did not find keywords in your abstract part. For example, you wrote “The work implements…”. But each article can only show how to implement something.

(2) The introduction is very similar to a big abstract. As for me, the introduction should show the problem to be solved. 

(3) The authors must clarify what is new from previous publications. As for me, the novelty it is presented insufficiently substantiated.

(4What is the need and importance of the proposed method? What are the potential uses of your proposed method?

(5)   What are the objectives and contributions of this work? Please describe them clearly. The problem statement must be improved. I would recommend the authors to show a few main points of both contribution and problem statement. 

(6)   Please describe your idea clearly. In the current paper, it is very hard to understand the main idea. What are the differences between your approach and others?

(7)  From my opinion, further qualitative and quantitative analysis of theexperimental results is required. Also, please, show the possible ways of your approach development.  

(8)   There are some English grammar mistakes. Please, correct them.

The results of research look promising. But the manuscript must be reworkedDespite the mentioned shortcomings, I consider the manuscript of good quality and deserving publishing after comments mentioned in the review have been addressed. Please note that practically all comments are of qualitative nature.

Comments on the Quality of English Language

Reviewer 2 Report

Comments and Suggestions for Authors

In this report, two FPGA implementations of a stable source identification algorithm in a circular, non-homogeneous region expressed in Fourier series expansions have been carried out. Overall, this manuscript is well organized. However, in order to further improve the quality of this manuscript, the following comments should be carefully considered:

1. The content in the abstract should be further condensed to highlight the innovative points of this manuscript. In introduction, the research background should be further strengthened and the descriptions of existing research should be further summarized.

2. In order to make it easier for readers to grasp the key points, the contributions should be further refined and listed point by point.

3. Please note whether the horizontal and vertical coordinates of all images need to be labeled with corresponding units.

4. From the references, it can be seen that further summary is needed regarding the research in recent years.

Comments on the Quality of English Language

Minor editing of English language required.

Reviewer 3 Report

Comments and Suggestions for Authors

The manuscript presents different problems. First off all, Abstract and Introduction must be rewritten. Both Abstract and Introduction have problems of English. There are too many abbreviations in this paper. Please fix these issues. In particular, no abbreviation must be present in the Abstract. More in general, the English grammar must be improved in order to improve the style of the paper. In the current version, the paper cannot be published. Secondly, the bibliography is incomplete. In particular, the references about fractal-wavelet modeling are not present. Please add a short citation in Introduction about the fractal-wavelet techniques. Therefore, I suggest adding the references below. 1. A Theory for Multiresolution Signal Decomposition: The Wavelet Representation. IEEE Transactions on Pattern Analysis and Machine Intelligence, 11, 674-693. 2. Chebyshev wavelet analysis, Journal of Function Spaces, 2022(1), 5542054, 2022. 3. Hyperspectral image classification using wavelet transform-based smooth ordering, Int. J. Wavelets Multiresolut. Inf. Process, 17(6), Article Number: 1950050, 2019. 4. Harmonic Sierpinski Gasket and Applications, Entropy, 20(9), 714, 2018. 5. A Framework of Adaptive Multiscale Wavelet Decomposition for Signals on Undirected Graphs, IEEE Transactions on Signal Processing, Volume: 67, Issue: 7, Pages: 1696-1711, 2019. 6. Primality, Fractality and Image Analysis, Entropy, 21(3), 304, 2019. 7. Fractional-Wavelet Analysis of Positive definite Distributions and Wavelets on D'(C), in Engineering Mathematics II, Silvestrov, Rancic (Eds.), Springer, pp. 337-353.

Comments on the Quality of English Language

The paper needs a full and comprehensive English review.

Round 2

Reviewer 1 Report

Comments and Suggestions for Authors

I have analyzed the answers of authors. I think the authors have corrected the article according with my remarks. 
The article is interesting for readers.
I propose to accept the paper in its current form

Comments on the Quality of English Language

No correction 

Reviewer 3 Report

Comments and Suggestions for Authors

The paper needs a major revision. In particular, English and References are weak. For the language, we suggest a review made by a native English speaker. For References, I invite the authors to cite recent results in fractal-wavelet analysis. Thus, I suggest adding the following references (or other ones of the same scientific weight, in accordance with the current MDPI policy).

1. Chebyshev wavelet analysis, Journal of Function Spaces, 2022(1), 5542054, 2022

2. Fractional-Wavelet Analysis of Positive definite Distributions and Wavelets on D'(C), in Engineering Mathematics II, Silvestrov, Rancic (Eds.), Springer, pp. 337-353,2016.

3. Hyperspectral image classification using wavelet transform-based smooth ordering, Int. J. Wavelets Multiresolut. Inf. Process, 17(6), Article Number: 1950050,2019.

4. On the Weierstrass-Mandelbrot fractal function, Proc. R. Soc. Lond., Ser. A, vol. 370, no. 1743, pp. 459-484, 1980.

5. A Framework of Adaptive Multiscale Wavelet Decomposition for Signals on Undirected Graphs, IEEE Transactions on Signal Processing, Volume: 67, Issue: 7,Pages: 1696-1711, 2019.

Comments on the Quality of English Language

The paper needs an extensive revision for the academic English.
